# "People here live in denial": A qualitative study of the pervasive impact of stigma on asthma diagnosis and care in Kenya and Sudan

Rana Ahmed[1‡], Stephen Mulupi[2,3‡], Miriam Taegtmeyer[4], Jane Ardrey[3], Graham Devereux[4], Martha Chinouya[5], Rashid Osman[1], ElHafiz Hussein[1], Sundos Modawey[1], Hoyam Eltahir[1], Caroline Waithera[2], Helen Meme[2], Elizabeth H. Shayo[6], Asma El Sony[1], Rachel Tolhurst[3*], IMPALA consortium[7]

1 Department of Lung Health, The Epidemiological Laboratory, Khartoum, Sudan, 2 Kenya Medical Research Institute, Centre for Respiratory Diseases Research, Nairobi, Kenya, 3 Department of International Public Health, Liverpool School of Tropical Medicine, Pembroke Place, Liverpool, United Kingdom, 4 Department of Clinical Sciences, Liverpool School of Tropical Medicine, Pembroke Place, Liverpool, United Kingdom, 5 Faculty of Education, Liverpool School of Tropical Medicine, Pembroke Place, Liverpool, United Kingdom, 6 National Institute for Medical Research, Barack Obama Drive, Dar Es Salaam, Tanzania, 7 Emmanuel Addo-Yobo, Brian Allwood, Hastings Banda, Imelda Bates, Amsalu Bingedie, Adegoke Falade, Jahangir Khan, Maia Lesosky, Bertrand Mbactcho, Hellen Meme, Kevin Mortimer, Beatrice Mutayoba, Louis Niessen, Jamie Rylance, William Worodria, Heather Zar, Elijah Zulu, Jeramiah Chakaya, Lindsay Zurba, S Bertel Squire

‡ Joint first authors.

* Rachel.Tolhurst@lstmed.ac.uk

## Abstract

Epidemiological studies show a global increase in asthma, straining healthcare systems in low- and middle-income countries. There are multiple barriers to accessing diagnosis and treatment in Sub-Saharan African countries like Kenya and Sudan such as healthcare infrastructure, diagnostic tools, healthcare workers' capacities, and cost. Asthma can be well controlled using safe and cost-effective treatments such as inhalers. Stigma related to asthma negatively impacts treatment-seeking and adherence in higher-income settings, with limited information about such impacts in Sub-Saharan Africa. We conducted qualitative interviews and Focus Group Discussions in Kenya and Sudan to explore health systems aspects of diagnosis and management of chronic respiratory diseases. Participants included patients, primary care healthcare workers, hospitals, and community actors. Data were analysed through a framework approach; our initial analysis showed that asthma stigma was prevalent in both countries. Further analysis was using the Health, Stigma, and Discrimination Framework by Stangl. Negative perceptions about the aetiology and prognosis of asthma contribute to stigma. Anticipated, internalized stigma, and enacted stigma affects individuals with asthma, encouraging them to hide their symptoms and resist diagnosis. This contributes to delayed healthcare seeking and treatment uptake, impacting both individuals with asthma and health professionals. Overall, stigma exacerbates challenges in communicating diagnosis, managing the illness, and maintaining psychosocial health and well-being for those with

Data availability statement: The data used in this study is stored on the LSTM repository and is publicly available at https://doi.org/10.57978/cbqt-hj69

Funding: This work is part of the NIHR Global Health Research Unit on Lung Health and TB in Africa at LSTM "IMPALA" program, which was commissioned by the National Institute of Health Research using Official Development Assistance (ODA) funding. Grant number 16/136/35. The funder played no role in the study design, data collection and analysis, decision to publish, or preparation of the manuscript. Salary costs of all authors were covered by this grant for their work on designing the study, collecting, analysing and reporting data and disseminating findings.

Competing interests: The authors have declared that no competing interests exist.

asthma. Poor asthma control can exacerbate fear and stigma. Improving asthma control has the potential to reduce fear and positively influence community norms. The rollout of inhalers and spacers for asthma treatment should be accompanied by deliberate stigma reduction strategies and awareness raising at all levels of the system.

## Introduction

### Asthma care

The noncommunicable lung disease asthma is the most common chronic disease of childhood, it is characterised by variable airflow obstruction [1]. Symptoms include wheeze, cough, breathlessness, and acute deteriorations in symptoms (exacerbations) requiring input from health services for treatment. Although asthma has an inherited component [2], there is no family history in 50% of cases and environmental exposures are important in aetiology. In sub-Saharan Africa (SSA) the prevalence of asthma is 9–20% [3]. Prevalence has increased in recent decades [4] with the highest prevalence of severe asthma in African children [5–7]. Childhood asthma is 'underdiagnosed and undertreated' in low-and-middle-income countries (LMICs) [8] resulting in high mortality and morbidity burden that includes adverse developmental outcomes, school absence, reduced academic achievement and social opportunities and (as adults) disadvantages in employment and livelihoods. Poorly controlled asthma is also a cause of catastrophic healthcare expenditure, exposing whole households to the risk of impoverishment [9].

Asthma diagnosis is based on clinical evaluation of symptoms and a therapeutic trial of treatment [10]. When available a clinical diagnosis may be supported by tests such as spirometry, peak flow monitoring, and exhaled nitric oxide measurement. Asthma diagnosis and management is difficult in sub-Saharan African (SSA) contexts where primary care is poorly resourced, healthcare workers lack training in asthma diagnosis and systems are not designed for chronic disease management [11]. There is no cure for asthma; treatment aims to control symptoms and reduce exacerbations. Typical asthma treatment comprises a combination of 'preventer' and 'reliever' therapies, most effectively administered by inhaler via a spacer device if a metered dose inhaler is used. These treatments are of proven efficacy, safe and highly cost-effective, reducing the incidence of acute exacerbations by 70% and making asthma deaths largely preventable [12]. Although these inhalers are on the WHO essential medicines list, spacers are not.

To date, efforts to address gaps in asthma care in SSA have focused on improving 'supply side' factors such as policy/guideline documents, training healthcare workers and improving access to peak flow meters and inhaled treatments. For example, Epilab (a local health non-governmental organisation in Sudan) introduced an asthma standard case management programme to Gezira State district hospitals, which saw a significant reduction in hospital, and emergency-room admissions [13]. Unfortunately, this is the exception rather than the rule, as most initiatives to improve availability and affordability have been unsuccessful [14,15]. Notably, the provision

of low-cost inhalers by the Asthma Drug Facility failed because of a lack of demand, despite a high burden of asthma and drastic reductions in asthma-related emergencies among those who did use provided inhalers [14]. Most interventions have failed to address 'demand side' factors (socio-economic context, perceptions and behaviours of individuals, households and communities), which are important in determining how and when health services are accessed. Uptake may continue to be poor when the perceived risks to the individual outweigh the benefit of health seeking.

## Stigma

Stigma is increasingly recognised as an important determinant of asthma control in the global north [16–20], where it is associated with delayed diagnosis, reluctance to use inhalers and reduced quality of life [21]. However, little is known about asthma stigma in SSA. In Tanzania, children fear eating and playing with children with asthma [22]; in Sudan widespread asthma stigma particularly affects young women, impacting marriage prospects and quality of life [23]. In Uganda respiratory symptoms are stigmatised because of their link with TB and with asthma thought to be contagious;adults try to keep their child's wheezing secret [24]. In Kenya, only 33% of parents of children with asthma accepted the diagnosis, 34% feared inhalers and only 31% felt that long term therapy was necessary [25]. While addressing knowledge gaps is important [26], a deeper understanding of how stigma undermines asthma care is needed in the context of SSA.

Stigma is a contested concept, influentially defined by Goffman as an "attribute that is deeply discrediting" [27]. Jones et al. (1984) [28] build on this to define stigma as a "mark" (attribute), linking a person to undesirable characteristics (or stereotypes). This focus on stigmatised individuals has been critiqued, in favour of understanding stigma as a process occurring through social and structural forces across multiple socio-ecological levels [29–31], emphasising an understanding of stigma in the context of social determinants of health, such as poverty and social stratification [32,33].

Arguing against constructions of stigma focused on specific diseases, Stangl et al (2019) [31] propose a framework drawn from stigmatised health conditions (including HIV, mental health, and obesity) to inform research, intervention development, and policy on health-related stigmas, which we apply in this paper. They identify four key domains of the stigma process, which ultimately create health and social impacts: 1) drivers and facilitators of stigma; 2) 'stigma marking' (also known as labelling) [29], which is influenced by gender, race, occupation, ethnicity, or kinship; 3) stigma experiences (lived realities) and practices (beliefs, attitudes and actions of others towards a stigmatised person); and 4) outcomes for affected populations (including access to and uptake of healthcare) and relevant institutions (including health services) [29,34,35].

We present findings from two linked studies (one in Sudan and one in Kenya) investigating health system readiness to respond to people with symptoms of chronic respiratory disease (CRDs), including asthma. Although not primarily designed to study the impact of asthma stigma, it emerged as such a strong theme in both countries and across all participant types that we feel obliged to report these findings separately in the hope they can assist programmes responding to the increasing need for asthma care in SSA. This paper adds to the evidence generated in earlier studies in Kenya, Sudan and Tanzania, highlighting policy and health care-level impediments to effective diagnosis and management of asthma in East African healthcare facilities [36,37].

## Materials and methods

### Ethics statement

The study was approved by the Research Ethics Committees of Liverpool School of Tropical Medicine (refs18–043; 19–003), the Ministry of Health of Gezira state, Sudan (ref 44/T/KH/1) and the Kenya Medical Research Institute (KEMRI) Scientific and Ethics Review Unit (ref SERU No. 3848). We obtained permission from district authorities, including by the county government of Meru (Ref. MRU/MED/MRU/C.50). We obtained written informed consent from all participants. Additional consent was obtained from parents for the adolescent FGD group.

**IMPALA (parent study) overview and sub-study designs**

We conducted two linked studies within the NIHR-funded International Multidisciplinary Program to Address Lung Disease and Tuberculosis in Africa (IMPALA) [38], an 11-country consortium aiming to improve the health of children and adults in Africa through multi-disciplinary applied health research on lung health and TB. The first study, in Sudan, used an action research design. An initial health systems analysis and applied qualitative methods underpinned by an interpretivist epistemology [39] engaged actors from within healthcare facilities and community subsystems to collate experiences, identify requirements and generate context-specific interventions to integrate chronic lung disease services into existing health systems. The second study, in Kenya, investigated public healthcare facilities' readiness to respond to people presenting with symptoms of chronic respiratory diseases, within the context of a devolved county government system. Both studies explored community members' perceptions of chronic respiratory diseases, asthma and care seeking experiences, the roles of community actors (including community health volunteers (CHVs) in Kenya) in supporting referrals to healthcare facilities, and healthcare workers' experiences in diagnosing, treatment and subsequent management of asthma. We use the COREQ reporting guidance to present our methods.

**Settings**

In Sudan, the study was based on eight purposively selected health facilities and their catchment communities in two localities within Gezira State. We selected Gezira State, because there was an asthma management program at district level hospitals where the Comprehensive Approach to improve quality of Lung Health Services (CLHS) has been implemented by EpiLab, [13,40] a local health research and development NGO with a mission to create and maintain access to effective health services. We purposively selected health facilities to represent experiences from district and health centre level facilities, in urban and rural locations, including those with and without support from Epilab. Four health facilities with Epilab support had an emergency room with nebulizers, and oxygen supply/canisters, an asthma clinic with a peak flow meter, a medical officer, a nurse and clerk or medical assistant for recording and reporting, while others only had emergency rooms with nebulizers, and oxygen supply/canisters.

In Kenya, the study was conducted in Meru County, central Kenya. We purposively selected four rural public healthcare facilities - two health centres, two primary referral hospitals, distributed across two of the nine sub-counties within Meru County. The fifth facility, Meru Teaching and Referral Hospital, was selected as it is a regional tertiary hospital, and we wanted to explore referral pathways. All the health facilities provided diagnostic and treatment services, for asthma. None of the public facilities provided spirometry services, although peak flow meters were mentioned as available at hospital level. Similarly, only hospitals had medical officers.

**Selection and recruitment of participants.** Study participants in both contexts were selected purposively, for their roles as (potential) users and consumers of healthcare services from the local healthcare facilities (community members and patients); or support in the provision of healthcare services (community health volunteers and facility healthcare workers). All participants were adults (over 18) and provided written informed consent (Table 1).

**Data collection.** Data were collected in Kenya between 16/07/2019–30/11/2020, with a pause due to COVID-19 restrictions (between 19/03/2020 and 15/07/2020), and in Sudan between 01/02/2019–31/08/2019. All interviews and focus group discussions (FGDs) were conducted by research assistants trained in qualitative research methods, who introduced themselves as researchers from their respective institutions and explained their interest in lung health with a view to improving diagnosis and management.

FGDs were conducted in the community at venues preferred by the participants. Health workers' interviews were conducted within the health facility premises.

In Kenya, interviews were led by a single, male investigator (SM, MSc) and four research assistants (1 male (DM), 3 females (RA, NC, ES)) who also took field notes on key issues of discussion. Interviews involving healthcare workers were carried out in English, whilst interviews with CHVs and community members were conducted in Swahili. In Sudan, all

**Table 1. Comparison of participant selection and recruitment.**

| Study participants | Sudan | Kenya |
|---|---|---|
| Primary care and hospital clinicians | Purposively selected to include different types of clinical role (general practitioners, medical director and medical assistants) who had a wide range of years of experience in clinical work and administration | Purposively selected based on their roles in providing care to patients with symptoms of chronic respiratory diseases, and included clinical officers, medical doctors, medical social workers. |
| Community actors | Purposively selected from the general community in areas served by the selected health facilities to reflect a range of literacy levels, socio-economic backgrounds, distance (near or further) from the health facilities, and experiences of living with, or being affected by asthma. | All Community Health Volunteers (CHVs) supporting eight Community Health Units (CHU) linked to the primary health facilities and hospitals. |
| Male and female asthma patients* | Purposively selected to represent all asthma severities and catchment communities at varied distances from the health facility; identified by the field team and health workers from the Asthma Standard Case Management Programme at study sites. | Selected through convenience sampling, based on attendance at study site during the study period with asthma confirmed clinically during same day consultation (including new and former diagnosis) |

*Clinically confirmed asthma.

interviews were conducted in Arabic and led by three male co-investigators (RK, MBBS, MSc; EH, BSc; BN, MBBS, MSc) and three research assistants (1 male and 2 females) (SA, HE, SG).

**Health worker interviews:** Health workers' interviews in both contexts were guided by semi-structured topic guides (see S1 Text, S2 Text) exploring health workers' perceptions of health facility readiness for integration of CRD management, with a focus on asthma; perceptions, expectations and threats for anticipated integration of CRDs.

**Community Focus Group Discussions (FGDs):** In Sudan, community FGDs were organised by gender and age (younger (18–39 years) and older (≥40 years)) in line with the social structure and norms of the local Sudanese society. A separate focus group of young women (18–39) was also conducted to explore the observation that asthma is a highly stigmatising illness in this group in Sudanese society. FGDs explored community understanding of CRDs especially asthma, community care-seeking practices for CRDs, and community experiences with and priorities for care, including risks and economic impacts encountered.

In Kenya, we conducted focus group discussions with CHVs supporting eight Community Health Units (CHUs) that were linked to the primary health facilities and hospitals. In each CHU, FGDs were stratified by gender and consisted of 6–8 participants. Topic guides for CHVs examined their understanding of asthma, roles and experiences in providing care (including supporting referrals of community members to healthcare facilities for CHVs), and perceptions of stigma for asthma in the community. Additionally, FGDs were conducted with community members living within the catchment areas of five CHUs. These were further stratified by age (aged <35 years, and >35 years, to represent younger and older participants respectively). Most of the community FGDs were conducted in one sub-county; a government ban on community meetings to control the spread of COVID-19 meant that we could not hold FGDs in the second sub-county. FGDs explored community members' perceptions about CRDs, asthma healthcare-seeking practices and experiences with asthma management services.

Interviews and FGDs were audio-recorded using digital audio recorders, transferred to laptops the same day of the interview and then transcribed using the denaturalised approach, in the original interview language using MS Word, by research assistants to produce verbatim transcripts [41]. Transcripts were translated and back translated from Arabic/Swahili by professional translators and reviewed for meaning. Transcripts were compared with field notes to triangulate data, add supplementary information and enrich the transcripts. Due to distances to data collection sites and literacy levels it was not possible to check transcripts with participants.

**Data analysis.** Initial thematic analysis of the data was undertaken separately in the two countries using the framework approach [42], which revealed stigma as an important theme. We then developed a coding framework based on Stangl's 2019 framework. [31] See Fig 1.

Key domains described in the Stangl framework were applied in the analysis: 1) drivers and facilitators of stigma are those influences that encourage, fuel, or shape stigmatising attitudes; 2) an outcome of drivers/ facilitators, 'stigma marking' implies an application of stigma which is shaped by personal characteristics such as gender, race, occupation, ethnicity, and kinship; 3) consequently, stigma experiences are observed, i.e., lived realities and practices based on attitudes, such as discrimination, avoidance, verbal abuse and other harms; 4) similarly, self/ internalised stigma may occur, i.e., "a stigmatized group member's own adoption of negative societal beliefs and feelings, as well as the social devaluation, associated with their stigmatized status" [34,35]; and 5) finally, stigma practices include attitudes, prejudices and behaviours of other people towards a stigmatised person(s). We further distinguish between perceived stigma (i.e., perceptions about how stigmatised groups are treated in a given context); anticipated stigma (i.e., expectations of bias being perpetrated by others) and [43] the 'enacted stigma' of overt acts of social exclusion [43].

Country research team members independently applied the Stangl framework [31] to develop the overarching themes. The final common coding framework was agreed after reviewing transcripts together and resolving any discrepancies. Separate analytical frameworks were developed for health worker interviews, patient interviews and community discussions. Codes were organised into categories and themes for analysis by charting into an Excel spreadsheet. Emerging themes were compared across interview categories. Due to distances to and between data collection sites it was not possible to conduct participant checking of themes.

**Participant characteristics.** Table 2 below summarises the characteristics of the study participants in Kenya and Sudan.

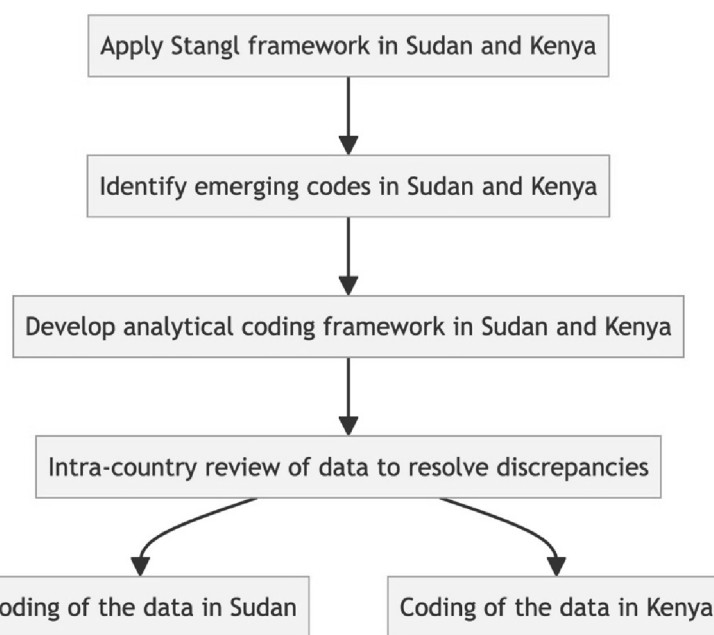

**Fig 1. Coding framework development across Kenya and Sudan.**

PLOS Global Public Health

**Table 2. Summary of participant characteristics.**

| Participant group | Sudan | | Kenya | |
| --- | --- | --- | --- | --- |
| | No and methods | Description | No and methods | Description |
| Primary care and hospital clinicians | 4 In-Depth Interviews (IDIs)+ | 2 general practitioners<br>1 medical director<br>1 medical assistant<br>2 males and 2 females<br>with 1–20 years of experience | 13 IDIs | 6 clinical officers<br>2 medical officers<br>1 physician; 4 pharmacy staff<br>(5 females and 8 males) |
| Community actors | 6 FGDs | 1 adolescent girls group (15–17 years)<br>1 females group ≥40 years<br>2 males groups ≥40 years<br>2 males groups < 40 years | 8 FGDs, 6 IDIs | 4 FGDs- community members:<br>- Adult community members >35 years, (1 male, 1 female)<br>-Young community members age 20–30 years, (1 male, 1 Female)<br>4 FGDs, female CHV<br>6 IDIs, (4 male CHV) |
| Male and female asthma patients‾ | 5 IDIs‾ | -4 females<br>-1 male<br>Illiterate to university level with age varies from 16-65 years old | 4 IDIs | 2 adult males >18 years old<br>2 adult females >18 years old |

+Clinicians with 1–20 years of experience.

‾Illiterate to university level with age varies from 16-65 years old.

## Results

Asthma is a stigmatised disease in both Sudan and Kenya. Some of the stigma domains of the Stangl framework however showed contextual variations across the two countries. This section highlights areas of convergence as well as divergent perspectives of the research participants across the two countries. Fig 2 shows the key findings with regard to four key domains identified by Stangl et al. (2019) [31]. These domains are inter-related in multiple directions; for example, stigma contributes to poor treatment outcomes, which in turn perpetuate the negative perceptions of people with asthma that shape resistance to diagnosis and manifestations of stigma.

### Theme 1. Drivers and facilitators

Varying perceptions were expressed about the aetiology of asthma. A common perception in both countries was that asthma is hereditary or 'genetic'. People with this view sometimes saw the aetiology as multi-factorial, acknowledging the relevance of other more immediate causes. For example,

> "We noticed these things many times that there are causing agents, but inheritance plays a very huge role. This is what I put my concentration on." (Male Community FGD, Sudan)

Perceiving asthma as hereditary often meant that it was also perceived as incurable.

> "We normally see asthmatic people in the community, and we view it as an inherited disease; it is there for life". (Male CHVs FGD, Kenya)

Another common view, particularly in Kenya, but also present in Sudan, was that asthma is infectious and can be transmitted to others.

> "They do say that if someone for example has asthma and they cough next to you and you are eating from the same plate, you would also be infected" (Female CHV IDI, Kenya).

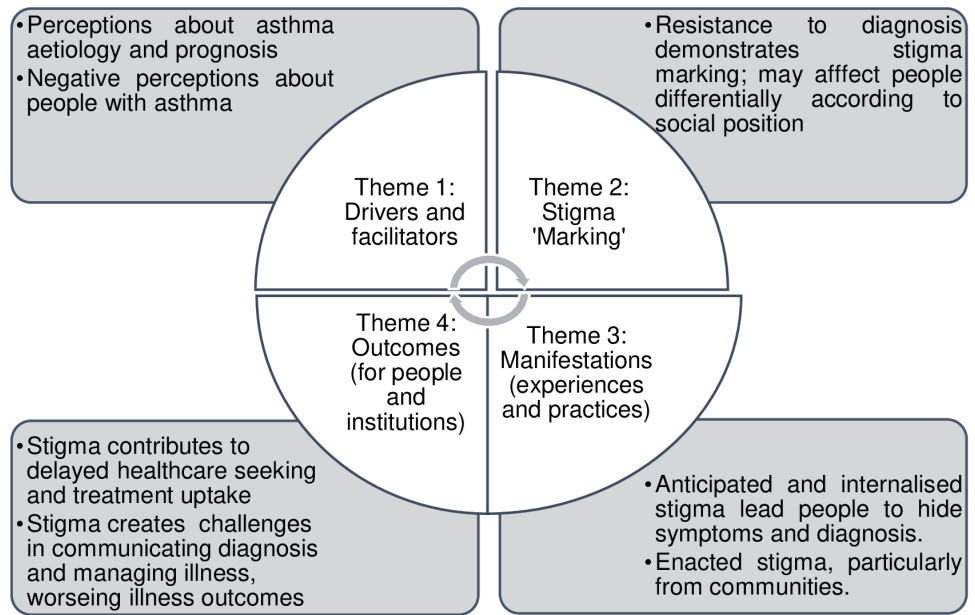

**Fig 2. Conceptual framework showing key findings.**

The persistent coughing seen by many as a sign of asthma was associated with tuberculosis, which is a stigmatised disease in both countries. In Kenya, symptoms of asthma such as difficulty in breathing were also associated with other stigmatised chronic conditions such as epilepsy as well as sometimes being attributed to witchcraft and/or seen as hereditary.

*"Some, especially those with little education, view those infected as having been bewitched or are victims of generational curses"* (Male CHV IDI, Kenya).

Fear of infection is therefore one of the drivers of both internalised and enacted stigma, which is discussed further in manifestations.

Negative perceptions commonly expressed about the capacities and social and economic situations of people with asthma highlighted that this is a health condition to fear. In both countries, people with asthma were perceived as 'weak' and unable to contribute economically or socially towards the family or community. For example, in FGDs in Sudan, a male participant commented, *"The patient will be transformed from a productive person to a consumer"* (Male Community FGD, Sudan), whilst women agreed *"He won't be able to work. That will affect him, and then it will affect the society by causing reduction in workers"* (Female Community FGD, Sudan). Asthma was perceived as reducing the range of livelihood opportunities open to people. For example, asthma symptoms were perceived to affect energy to do 'hard' physical work, which might expose them to dust or changes in temperature.

*" His opportunities are limited and if he is exposed to hard physical work like dust or air, coming in and out from warmth to cold … that will decrease the patient's job opportunities"* (Male Community FGD, Sudan)

Educational prospects were perceived to be affected:

*"My daughter is in high school level. She said there is a girl who gets attacks during school... She would like to study but her health condition will not allow. This will affect her performance results. It will absolutely affect her work later on."* (Female Community FGD, Sudan)

In both countries some people also perceived that asthma would affect the sexual performance of sufferers: "*Another thing about asthma, even the other 'work' in bed (sex) is hard*" (Female CHV FGD, Kenya).

Some participants added that people with asthma are not able to look after themselves. These perceived limitations on the capacities of people with asthma meant that they were viewed as a social burden.

In Kenya, perceptions of people with asthma as a burden to their families and communities were related to the anticipation that they would incur healthcare expenditure, including costs of travel to healthcare facilities during asthma emergencies:

*"People feel that this asthmatic person… will consume their money. Sometimes people with asthma are rushed to hospital at night, they feel the money they are spending on transporting them to hospital is a burden"* (Female CHV FGD, Kenya).

### Theme 2: Stigma marking

Peoples' reactions to a diagnosis of asthma demonstrated stigma 'marking'. In both countries, a range of participants described how many people responded by refusing the 'label' of asthma. For example, health workers in Kenya reported:

*"When you tell them it's asthma they say, 'No daktari [doctor]! No! Don't write asthma, in our family there is nobody who suffers from asthma, that's a bad illness, I don't want to be associated with it'"* (Clinical Officer IDI, Kenya).

In Sudan, people preferred the term 'allergy' instead of asthma:

*"Every person here has something making them fear the word "asthma", they call it* hassasya *(allergy)* (Male HW, IDI, Sudan)

There were mixed opinions about whether stigma marking affected people differently according to socio-economic status. In Kenya, one CHV perceived people with less education were more likely to attribute asthma to generational curses or witchcraft, whilst a healthcare worker emphasised resistance to asthma diagnosis was common regardless of educational background.

*"There is also the stigma attached to diseases like asthma. So, in my experience every time I tell a mum - even one who is educated - that your child could be having asthma, they tell you "No, where did he get it?"* (HW IDI, Kenya).

A patient in Kenya commented on the greater effect of asthma on people living in poverty due to lack of opportunity to avoid triggers:

*"It affects mostly the poor, do you know that? Because such poor people have to work whether it is cold or not- and they may get sick, but do not have anyone to support them* (Female Patient IDI, Kenya).

### Theme 3: Stigma manifestations (experiences and practices)

In both countries, most people with asthma diagnosis and/or symptoms reported hiding these, particularly from their communities. This included making efforts to appear healthy, suppressing their coughs, even when their airways were irritated and carrying on with activities as normal despite feeling unwell. For example,

*"I don't tell my friend clearly that I am irritated, and moreover; I am not showing that I have this disease."* (Female Patient, IDI, Sudan)

In Sudan most interviewees only disclosed symptoms to close family members, such as mothers, sisters, wives and brothers. One interviewee had told no one:

*"Not even anyone at home. As they say, the pain is all mine! Who could feel my pain?! They are our wounds, and we tolerate them alone (laughs)"* (Female Patient, IDI, Sudan)

Such concealment in both contexts related to both anticipated and internalised stigma. Interviewees shared how they wanted to prevent others worrying about them and how they feared no-one would help them, showing concern about being seen differently or discriminated against. They also explained they felt less able to fulfil expected roles in society. For example, one female interviewee in Sudan explained how she felt she was less useful to her family and community than previously:

*"I used to work in the fields, but I gave that up ever since I got sick. Now I just stay at home and cook for my kids"* (Female patient, IDI, Sudan)

Another female patient felt less able to contribute to community events: *"whenever I go to an occasion I'll only participate in the conversation, I can't help and serve as I used to."* (Female patient, IDI, Sudan)
A male patient similarly expressed that "*I can't do hard work; I can only do desk jobs or easy work…I can't handle exertion*" (Male patient, IDI, Sudan). Many interviewees' narratives suggest they have internalised negative societal perceptions of being a burden to others. In Kenya, some participants felt 'shame' and 'disgrace' leading them to avoid socialising and isolate themselves. As a health worker put it:

*"Some of the patients…pity themselves a lot, and they take it as if they are being burdens to others. So, they stigmatise even themselves" (Social Worker IDI, Kenya).*

In both countries, interview narratives showed people living with asthma could be discriminated against and socially isolated in their communities. Women in a focus group in Sudan agreed that people would generally avoid someone with a chronic cough for fear of infection: *"It is not nice - they avoid him/ her…. Everybody worries about himself)"* (Female Community, FGD, Sudan), whilst men expressed that chronic cough created vulnerability to theft or violence: "*He may get robbed…or anything else. Chest diseases may make a patient vulnerable to things like these…he may be tired and not able to defend himself*" (Male Community, FGD, Sudan). A CHV in Kenya stated that *"People will tend to move away saying you might infect them"* (Male CHV IDI, Kenya) and a female patient with asthma concurred: *"even neighbours wouldn't serve you a cup of tea, because people fear that you may infect them. They think it is like TB."* (Female Patient IDI Kenya)
In addition to fear of infection, in Kenya people with asthma were considered an inconvenience. Some community members excluded people with asthma from social events for fear that their illness may disrupt their plans.

*"Celebrations (laughs). There are those outings that we go to distant places, during the rainy season, you know, this asthmatic person is isolated or avoided, they won't go with them because they might get sick there and bring us complications. So, they are avoided (Female CHV FGD, Kenya)*

In both countries, discrimination against individuals with asthma and even their families was reported with regard to marriage proposals due to perceptions of heredity, which particularly affected young women in Sudan.

*"Asthma… will lead to single marital status: It will lower their market value"* (Female Community, FGD, Sudan).

"*So, if people know that that someone has asthma, they don't want their girls to get married to someone with asthma (Female CHV FGD, Kenya).

In Kenya, CHVs suggested their own lack of knowledge about asthma prior to training meant they avoided close contact with people with chronic cough, potentially exacerbating community fears:

*"But initially these diseases such as asthma and TB, you know [CHVs] would sometimes talk to [community members] from a distance because they feared contracting the same diseases. But now we have been trained to know"* (Male CHV IDI, Kenya)

Enacted stigma within families was less commonly reported. However, one asthma patient in Kenya narrated how her husband's family had been opposed to their marriage and encouraged him to reject her. Whilst her husband was initially supportive, she described how he later expressed regret due to the financial burden of the costs of her treatment:

*So … these problems caused financial strains and impoverished my husband. … government hospitals only offered drug prescriptions, but no medicines, so money was never sufficient for other household expenses. We couldn't even afford rent, so we had to move from our rental house to our rural home……now when we call his parents, they said "we told you to leave this woman alone, but you did not listen- this is a problem you have brought on yourself - asthma is not treatable"… It reached a point, I even despised myself…Even my husband… nowadays regrets - he tells me…that if I had disclosed my condition earlier that I have asthma, he wouldn't have married me"* (Female patient IDI, Kenya).

However, in both countries, a minority of community participants did not perceive any enacted stigma. For example, in Sudan some men in a focus group perceived that there were no social consequences for people with asthma:

*"I think the social aspect has very tiny effect"* (Male Community, FGD, Sudan).

Only one patient interviewed in Kenya did not feel that he needed to hide his condition, whilst a patient in Sudan did not feel surprise or concern at his diagnosis, *"because it runs in the family."* (Male patient, IDI, Sudan).

### Theme 4: Outcomes

In both countries, people with asthma, community members and healthcare workers agreed anticipating and internalising asthma stigma influenced their healthcare seeking behaviour. This included not seeking formal healthcare due to fear of a diagnosis. A woman with symptoms of asthma in Sudan explained she had not sought healthcare for her symptoms because she feared a diagnosis of asthma and the visibility that using an inhaler would bring:

*"Because I fear they may tell me that I have asthma and I have to use a spray (inhaler)"* (Female patient, IDI, Sudan)

Some people opted for home remedies, buying medicines from pharmacies and/or traditional treatments because of a perception they are suffering from 'allergies.' One patient diagnosed with asthma described how this delayed her initiation of treatment at hospital:

*"My family would say I should go to the doctor, (but) some of them say give her warm water, others say sesame oil and some say acacia plant. I would use them for three days, staying up the whole time until I feel better, but when I came here to Gezira I started going to hospitals and getting oxygen"* (Female patient, IDI, Sudan).

Women in a focus group explicitly related people hiding their symptoms and not seeking healthcare due to anticipated stigma:

*"Society's fear of the disease makes TB patients or patients with cough hide their illness and do not seek medical advice …so that people will not avoid them"* (Female community, FGD, Sudan)

In Kenya, concerns about the costs of care-seeking and the shame associated with asking for help were particularly emphasised.

*"Some people will not seek for help until their condition is so bad- they feel ashamed to ask for financial support from other people"* (HW IDI, Kenya).

In both countries, healthcare workers also reported that patients minimise the severity of their symptoms to avoid a diagnosis of asthma, in addition to delaying care seeking.

*"People here live in denial. One comes to you and says he has cough for 3 days, while the truth that he has cough for more than a year"* (Male HW, IDI, Sudan).

*"A patient comes in, you try to take her history- you ask, her do you cough? She says 'no'.. but as you continue the examination, she starts coughing- you ask, "what is that?" (laughter)"* (HW IDI, Kenya).

Healthcare workers in both countries expressed the challenges they faced with patients accepting a diagnosis of asthma, which contributes towards limited adherence to treatment.

*"First, the patient must be convinced that s/he has asthma in order to be managed since most of the patients over here are not convinced. They say they have allergy."* (Female HW, IDI, Sudan)

Resistance to a diagnosis of asthma affected patient willingness to use recommended treatments, including inhalers, oxygen and hospital admissions, and referral. For example, in Sudan, health workers explained some patients even refused to have oxygen when indicated due to a belief that oxygen therapy could convert a temporary allergy into a permanent condition of asthma:

*'They refuse because they believe that oxygen will make it permanent'.* The health worker later continued: *If you prescribe an inhaler, the patient will absolutely refuse to take the inhaler"* (Female HW, IDI, Sudan).

One reason for this was explained in community focus groups where the group discussed concerns about using an inhaler in public making the condition visible. Patients may instead purchase antihistamines or corticosteroids from pharmacies as treatment for 'allergies',

*"I have told you that no one will accept asthma diagnosis… People get accustomed to purchasing antihistamine; they know what they shall purchase from the pharmacy… Prednisone is constantly taken. The patient takes prednisone and never takes Ventolin. They call it allergy tablets. I get confused; I thought the patient was taking Antihistamine, and the patient showed me a prednisone strip. They take these tablets. Most patients do that"* (Female HW, IDI, Sudan).

Communication between health workers and patients can be negatively affected by their different understanding of 'allergy' treatments. Monitoring and evaluation of patients' progress or otherwise can also be negatively affected by patients' reluctance to engage with longer-term management:

*"However, the answer is always denial of asthma diagnosis and stating it is only allergy, even those cases presenting at the emergency department with attacks at night. We even tried to have a register book for those patients. Unfortunately, the patient leaves before obtaining any information; even we could not get the name of the patient"* (Male HW, IDI, Sudan).

In Kenya, health workers reported patients were reluctant to use spacers as well as inhalers, which particularly affected management of children and presented barriers to demonstration of inhaler use. Furthermore, open spaces for drug dispensation at the pharmacies in all facilities limit audio privacy, and appropriate communication of inhaler use. Some healthcare workers observed that patients would communicate about asthma in hushed tones when receiving medication so as not to be heard.

Health workers in both countries and community health volunteers (CHVs) in Kenya, observed stigma experiences and practices could exacerbate other co-morbidities, including mental health problems, diabetes and hypertension. For example, in Kenya CHVs related anticipated social isolation and anticipated stigma to the development of co-morbidities:

"*Now this person, since he is isolated, you know he develops bad thoughts; he feels that he is a threat to the community, and he can even develop diabetes or hypertension because of that. See now? you have made someone develop a disease they would not have developed*" (Female CHV FGD, Kenya).

However, CHVs felt their support and explanations could make a difference to stigma and isolation:

"*If you visit them and let's say the husband or wife has asthma you will see them very depressed. You see them in deep thoughts (and) self-pity. But if you tell them that the disease is just as normal as the others then they feel encouraged such that when you visit the next time she feels jovial*" (Female CHV FGD, Kenya).

In Sudan, health workers reported that patients hiding their illness could lead to deterioration and even death:

"*The health condition of some of those patients deteriorated as they hide their illness... They hide their illness till they die. Some patients did that*" (Female Community, FGD, Sudan).

## Discussion

Our findings contribute towards a limited literature on understanding stigma associated with asthma in SSA. We found that perceptions about aetiology and prognosis of asthma drive stigma in these contexts, including concerns about heritability, incurability and associations with other stigmatised diseases such as TB. People with asthma resist diagnosis as they fear a loss of social status and negative stereotypes, particularly a loss of capacity to fulfil economic and social roles in their families and communities. This stigma marking may affect people differentially according to their social position; such as limiting opportunities for marriage for young women in Sudan. Anticipated and internalised stigma are commonly reported by people living with asthma, while people with asthma and community members report enacted stigma. Anticipated, internalised and enacted stigma all contribute to delayed healthcare seeking and delayed (or limited) uptake of treatment, including self-medication (without appropriate information), giving incomplete histories to professionals, inappropriate treatments, refusing registration for chronic care, and repeated emergency care seeking. Stigma contributes towards challenges for health professionals in communicating diagnosis and managing illness, limiting effective management and contributing towards poor symptom control, poor mental health and poor psychosocial well-being.

Perceptions of asthma aetiology and prognosis are drivers of stigma and occur in specific socio-economic and health systems contexts. Health systems in many SSA countries are ill-prepared to diagnose and manage asthma [36,37,44], publicly available accurate information about the disease is limited, and services are financially and geographically inaccessible to many. In such contexts, this leads to a vicious cycle where high rates of acute exacerbations and death are observed due to uncontrolled asthma, exacerbating stigma and decreasing uptake of services. (Fig 3).

In contexts of low-income and insecure livelihoods, where livelihoods depend on physical labour, reduced capacities due to uncontrolled asthma are a threat to family economic security; educational prospects may also be affected, limiting

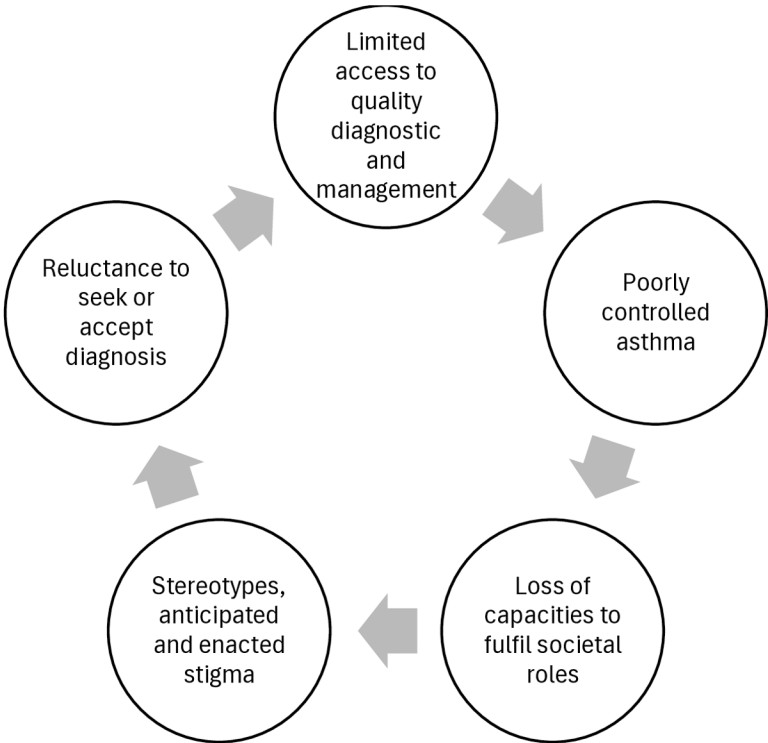

**Fig 3. The interplay of uncontrolled asthma management and stigma.**

future employment opportunities. Concerns about these economic and social threats may therefore be understood as 'situated rationality' [45]. Our research contributes to deepening understanding of asthma stigma as a social and structural process, rather than purely an interpersonal phenomenon.

## Perceptions of aetiology in health systems context

Perceptions of asthma aetiology identified in the study are partially in line with existing biomedical understandings. For example, perceptions of asthma as heritable are in line with evidence that around 50% of people with asthma have a family history of the disease [2]. However, a potential consequence of perceived heritability is that people with no known family history may not recognise the possibility that they have the condition. Whilst sometimes the perception of asthma as common within a family seems to support its acceptance, perceptions of heritability have differential consequences for young people, and especially women, in patriarchal societal contexts where marriage and childbearing are central to women's social and economic security [46]. Respiratory symptoms being equated with tuberculosis may result from widespread education on signs and symptoms by TB control programmes in SSA. As we have reported elsewhere, such programs rarely provide further diagnosis or care to individuals who test negative for tuberculosis [36,37]. Associations with TB lead to misplaced concerns about risks of infection from associating with symptomatic individuals. The conflation of asthma and 'allergies' is further in line with evidence of the role of atopy in asthma aetiology for some; in the Global North, about 50–60% of asthma in children has an allergic/atopic aetiology [47]. The term 'allergies' also appears to function as a label for symptoms that avoids the stigma marking associated with the diagnostic label of asthma. However, this has problematic consequences for treatment. Our study found that antihistamines and oral corticosteroids - e.g., prednisolone - were commonly used to treat symptoms which is consistent with a previous study [44]; these are not indicated in asthma

guidelines, and therefore risk wasting limited resources, causing side effects and further compounding perceptions that asthma cannot be effectively managed.

## Addressing stigma *through* demand and supply side approaches

Efforts to address gaps in asthma care in LMICs have focused on ensuring 'supply side' factors such as releasing new policy and guidelines, improving access to peak flow meters and drugs and training health care workers. Whilst these efforts are critically important to address the wider context facilitating stigma, our findings show engagement with the 'demand side' is important in determining how and when health services are accessed. In practice, demand and supply are interconnected and a holistic perspective is required, or uptake will continue to be poor as perceived threats to the individual outweigh the benefits of health seeking. Demand and supply particularly interact in clinical encounters between healthcare workers and patients, where anticipated stigma affects accurate history taking, poor explanation and negative patient responses (resistance) to asthma diagnosis. Addressing this challenge requires nuanced communication by health professionals in ways that engage with patients' perspectives and socio-economic realities as well as the biomedical evidence base. For example, it may be helpful to explain that whilst asthma may be inherited, this is not always the case and heritability need not be feared, since asthma is controllable with relatively low-cost medication. Similarly, the allergy label may potentially be useful if it provides patients with a way to access and use appropriate treatment, and they are informed that antihistamines are ineffective.

Learning from other stigmatised conditions reveals possible strategies. Experience with HIV diagnosis and treatment in SSA shows affordable access to medication, counselling and confidentiality contribute to stigma reduction, particularly because this results in adherence to anti-retroviral medication [48], which observably reduces or eliminates disease symptoms and prevents death. Successful treatment programs can lead to social diffusion of knowledge of positive outcomes, encouraging others to seek testing and treatment, breaking the vicious cycle seen in Fig 3. However, this has been achieved with substantial investments through vertical disease control programmes.

Tuberculosis (TB) control programs may similarly offer valuable insights for managing asthma in primary care settings due to the importance of medication adherence. Community Health Workers (CHWs) have played a vital role in TB programs by providing directly observed therapy (DOT) and building trust with patients [49]. Whilst training of health workers on both technical aspects of diagnosis and treatment and communication is important, limitations on time for patient consultation and privacy are likely barriers to this approach for asthma in general health services. The potential role of primary care and CHWs in asthma management requires careful consideration, considering the differences between the two conditions. Asthma does not require DOT, but ensuring long-term medication adherence remains crucial. Primary care physicians may provide personalised asthma management plans. CHWs may leverage their deep community connections, focusing on education, adherence support, and reducing stigma within communities. Public communication of this nuanced information will also be important to address misconceptions about asthma, highlight the benefits of treatment, and encourage open communication between patients and healthcare providers. By fostering understanding within families and communities, social diffusion of positive treatment outcomes can be harnessed to reduce stigma and encourage timely diagnosis and management of asthma.

Recent developments in asthma treatments such as the emphasis on the 'as needed' use of combined rapid onset inhaled long-acting beta-agonists with inhaled corticosteroids [50] are likely to support stigma reduction as treatment works quickly to improve symptoms [51] and have fewer side effects. Moreover, the prolonged duration of action of these inhaler preparations will also reduce the need for affected people to use an inhaler in public. Effective communication of this may also contribute to promoting adherence and reducing stigma. The Asthma Drug Facility initiative [14] found that where individuals with asthma who, were able to adhere to use of inhalers and attend follow-up, they experienced controlled asthma, and drastic reductions in asthma- related emergency visits and hospital admissions.

### Applying *a* stigma framework

Stangl's framework worked well for asthma stigma to distinguish between drivers of stigma (particularly regarding fear of the illness and its consequences), stigma marking (particularly relating to diagnosis and its communication), stigma experiences, and consequences (for people affected and for healthcare providers). We found it helpful to further distinguish stigma experiences and practices, for example between perceived, enacted and internalised experiences, since all were reported within our data and may have different implications for action. Our application of the framework enables holistic consideration of stigma as a socio-economically produced rather than individual interpersonal phenomenon, which requires actions across its dimensions by a range of actors at multiple socio-ecological levels, including policy, institutional and community.

### Limitations

The COVID-19 pandemic unfortunately hindered our ability to conduct all planned FGDs in Sudan and Kenya. This might have limited our ability to achieve saturation, the point at which no new significant information emerges from further data collection.

Moreover, since this was primarily a qualitative study focused on health systems, we did not collect data on the specific processes of asthma diagnosis or inhaler use by patients. This limits our ability to analyse opinions or comments regarding treatment outcomes, healthcare service utilisation related to asthma, or the patient experience with these aspects of their care. Since the study focused on adults, we could not fully explore the experiences of children. In addition, while stigma was not the original focus of the study, a comprehensive set of data emerged on this theme. However, due to the study design and focus on health systems, we may not have been able to fully explore the nuances of stigma associated with asthma.

### Conclusion

The underdiagnosis and undertreatment of asthma in the Global South is a human rights violation, since people with asthma continue to suffer and die despite availability of safe, highly clinically and cost-effective inhaled treatments widely used in the Global North. Our findings suggest action is required at multiple levels across the process of stigma production to create conditions that maximise the possibilities for patients to seek care, accept a diagnosis and adhere to highly effective inhaled medication and follow-up with the support of their families and communities. This approach offers the potential to intervene in the cycle of poor asthma management and stigma and create instead a virtuous cycle, in which experiences and observations of accessible, effective treatment significantly reduce the threat associated with the label of asthma.

### Supporting information

**S1 Text. Topic guides, Sudan.**
(DOCX)

**S2 Text. Topic guides, Kenya.**
(DOCX)

### Acknowledgments

We acknowledge the county government of Meru, healthcare workers and community members for supporting this research; and the research team members: Damian Mwaura, Noel Chemwa, Ruth Akiso, Elizabeth Sitati for supporting data collection in Kenya. We would also like to express our gratitude for the support received from the Ministry of Health

in Gezira State, the health workers at both the Epilab and Non-Epilab sites, the patients, the members of the community, and the research team, including Ms. Hana A.Elsadig, Dr. Bandar Noory, Hind Eltigani, Sayed Gamal (may his soul rest in peace), Sahar Aladdin, and Ayaa Abddin. Their assistance was invaluable in conducting research activities and collecting data in Sudan. The views expressed in this publication are those of the author(s) and not necessarily those of the NHS, the National Institute for Health Research or the Department of Health.

## Author contributions

**Conceptualization:** Rana Ahmed, Stephen Mulupi, Miriam Taegtmeyer, Jane Ardrey, Martha Chinouya, Rachel Tolhurst.

**Data curation:** Rana Ahmed, Stephen Mulupi, Rashid Osman, ElHafiz Hussein, Hoyam Eltahir, Caroline Waithera, Helen Meme.

**Formal analysis:** Rana Ahmed, Stephen Mulupi, Jane Ardrey, Sundos Modawey.

**Funding acquisition:** Miriam Taegtmeyer, Rachel Tolhurst.

**Investigation:** Miriam Taegtmeyer, Rachel Tolhurst.

**Methodology:** Rana Ahmed, Stephen Mulupi, Miriam Taegtmeyer, Jane Ardrey, Martha Chinouya.

**Project administration:** Rana Ahmed, Stephen Mulupi, Rashid Osman, ElHafiz Hussein, Asma El Sony.

**Supervision:** Miriam Taegtmeyer, Jane Ardrey, Martha Chinouya, Asma El Sony, Rachel Tolhurst.

**Validation:** Miriam Taegtmeyer, Jane Ardrey, Martha Chinouya, Rachel Tolhurst.

**Writing – original draft:** Rana Ahmed, Stephen Mulupi.

**Writing – review & editing:** Rana Ahmed, Stephen Mulupi, Miriam Taegtmeyer, Jane Ardrey, Graham Devereux, Martha Chinouya, Rashid Osman, ElHafiz Hussein, Sundos Modawey, Hoyam Eltahir, Caroline Waithera, Helen Meme, Elizabeth H Shayo, Asma El Sony, Rachel Tolhurst.

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
