## [Decision Letter · Decision Letter 0]

31 Mar 2025

PGPH-D-24-02269

“People here live in denial”: A qualitative study of the pervasive impact of stigma on asthma diagnosis and care in Kenya and Sudan

Dear Dr. Tolhurst,

Thank you for submitting your manuscript to PLOS Global Public Health. After careful consideration, we feel that it has merit but does not fully meet PLOS Global Public Health’s publication criteria as it currently stands. Therefore, we invite you to submit a revised version of the manuscript that addresses the points raised during the review process.

Comments from the Editorial office: Please note that we are issuing a decision on your manuscript at this point to prevent further delays in the evaluation of your manuscript. Please be aware that the editor who handles your revised manuscript might find it necessary to invite additional reviewers to assess this work once the revised manuscript is submitted. 

We look forward to receiving your revised manuscript.

Kind regards,

Annesha Sil, Ph.D.

Staff Editor

PLOS 

Journal Requirements:

2. Please make sure the funding information on the submission form matches your financial disclosure statement. Please indicate by return the full and correct funding information for your study and confirm the order in which funding contributions should appear. Please be sure to indicate whether the funders played any role in the study design, data collection and analysis, decision to publish, or preparation of the manuscript.

3. Please insert an Ethics Statement at the beginning of your Methods section, under a subheading 'Ethics Statement'.

4. In the online submission form, you indicated that “The data is owned by LSTM and is subject to legal restrictions on sharing. The data used in this study is stored on the LSTM repository. The data can be made available for the evaluation of reported analyses with a signed data access agreement. Data access requests may be sent to LSTM at Rachel.Tolhurst@lstmed.ac.uk.”. 

3. Uploaded as supplementary information.

Additional Editor Comments (if provided):

Reviewers' comments:

Reviewer's Responses to Questions

**Comments to the Author**

1. Does this manuscript meet PLOS Global Public Health’s publication criteria ? Is the manuscript technically sound, and do the data support the conclusions? The manuscript must describe methodologically and ethically rigorous research with conclusions that are appropriately drawn based on the data presented.

Reviewer #1: Yes

Reviewer #2: Yes

2. Has the statistical analysis been performed appropriately and rigorously?

Reviewer #1: Yes

Reviewer #2: N/A

3. Have the authors made all data underlying the findings in their manuscript fully available (please refer to the Data Availability Statement at the start of the manuscript PDF file)?

Reviewer #1: Yes

Reviewer #2: Yes

4. Is the manuscript presented in an intelligible fashion and written in standard English?

Reviewer #1: Yes

Reviewer #2: Yes

5. Review Comments to the Author

Reviewer #1: Thank you for the opportunity to review this manuscript. The purpose of the study was to examine the pervasive impact of stigma on asthma diagnosis and care in Kenya and Sudan via focus groups. These focus groups were conducted with patients, primary care providers, hospitals and community actors. The qualitative analyses were guided by Stangl’s Health, Stigma, and Discrimination Framework. There were a number of interesting findings revealed from these analyses that expand the construct of stigma beyond a personal attribute to a process which occurs through social and structural factors across multiple socio-ecological levels. The study was well-designed, methods were robust, and the analytical strategy rigorous. The findings from these focus groups reveal that perceptions about the etiology and prognosis of asthma drive stigma and behaviors around social status, employability, denial of asthma diagnosis, refusal to seek care, and lack of treatment or inappropriate treatment. Healthcare systems in SSA are not prepared to diagnose and manage asthma and when combined with patients’ reluctance to seek care or disclose complete histories, which leads to high rate of exacerbations and death. The authors did a nice job of placing these findings within the context of research on other chronic diseases such as HIV and TB and identified several potential solutions that must occur across multiple levels in order to reduce stigma, enhance diagnosis and access to effective medications, and reduce morbidity and mortality. There was also an acceptable discussion of study limitations.

Reviewer #2: The paper entitled ““People here live in denial”: A qualitative study of the pervasive impact of stigma on asthma diagnosis and care in Kenya and Sudan” describes interviews and focus groups in Kenya and Sudan exploring chronic respiratory diseases. The authors found that negative perceptions about asthma contribute to stigma, which can in turn contribute to delayed healthcare seeking and treatment uptake. The authors suggest that treatment should be accompanied by stigma reduction strategies and awareness raising. Overall, the paper highlights an important topic area. Minor suggestions for improvement are detailed below.

Abstract

Should read ‘This contributes…’

Introduction

The introduction section is quite long for the reader, and it would benefit from a reduction in word count to provide a more succinct overview of the topic area. In particular, the definition/descriptions of stigma could be reduced, such as the Stangl framework, as this is discussed in the methods section.

Methods

It would be useful if the manuscript makes reference to the adherence to the COREQ reporting guideline https://www.equator-network.org/reporting-guidelines/coreq/

Final row Table 1 – Kenya, should read ‘convenience sampling’.

Line 189: the sentence needs wording to make clear that those dates were the pause dates. Perhaps ..COVID-19 restrictions (between 19/02/2020 and 15/07/2020), and in Sudan…’

Are the topic guides available in the supplementary materials for focus group discussions?

Table 2 – IDI: acronym spell out first use.

Results

Figure 2 – Theme 3 – heading text ‘manifestations’ is across two lines.

Discussion

The discussion is also on the longer side and would benefit from a reduction in word count.

Lines 557-559 need a citation to back up this evidence.

There are minor typos and spacing typos in the manuscript.

6. PLOS authors have the option to publish the peer review history of their article (what does this mean? ). If published, this will include your full peer review and any attached files.

**Do you want your identity to be public for this peer review?** For information about this choice, including consent withdrawal, please see our Privacy Policy .

Reviewer #1: No

Reviewer #2: No

---

## [Decision Letter · Decision Letter 1]

21 Jul 2025

“People here live in denial”: A qualitative study of the pervasive impact of stigma on asthma diagnosis and care in Kenya and Sudan

PGPH-D-24-02269R1

Dear Dr. Tolhurst,

We are pleased to inform you that your manuscript '“People here live in denial”: A qualitative study of the pervasive impact of stigma on asthma diagnosis and care in Kenya and Sudan' has been provisionally accepted for publication in PLOS Global Public Health.

Best regards,

Julia Robinson

Executive Editor

Reviewer Comments (if any, and for reference):

Reviewer's Responses to Questions

**Comments to the Author**

1. If the authors have adequately addressed your comments raised in a previous round of review and you feel that this manuscript is now acceptable for publication, you may indicate that here to bypass the “Comments to the Author” section, enter your conflict of interest statement in the “Confidential to Editor” section, and submit your "Accept" recommendation.

Reviewer #1: All comments have been addressed

Reviewer #2: All comments have been addressed

2. Does this manuscript meet PLOS Global Public Health’s publication criteria ? Is the manuscript technically sound, and do the data support the conclusions? The manuscript must describe methodologically and ethically rigorous research with conclusions that are appropriately drawn based on the data presented.

Reviewer #1: Yes

Reviewer #2: (No Response)

3. Has the statistical analysis been performed appropriately and rigorously?

Reviewer #1: Yes

Reviewer #2: (No Response)

4. Have the authors made all data underlying the findings in their manuscript fully available (please refer to the Data Availability Statement at the start of the manuscript PDF file)?

Reviewer #1: Yes

Reviewer #2: (No Response)

5. Is the manuscript presented in an intelligible fashion and written in standard English?

Reviewer #1: Yes

Reviewer #2: (No Response)

6. Review Comments to the Author

Reviewer #1: Thank you for the opportunity to review a revision of this manuscript. The purpose of the study was to examine the pervasive impact of stigma on asthma diagnosis and care in Kenya and Sudan via focus groups. These focus groups were conducted with patients, primary care providers, hospitals and community actors. The qualitative analyses were guided by Stangl’s Health, Stigma, and Discrimination Framework. There were a number of interesting findings revealed from these analyses that expand the construct of stigma beyond a personal attribute to a process which occurs through social and structural factors across multiple socio-ecological levels. The study was well-designed, methods were robust, and the analytical strategy rigorous. The findings from these focus groups reveal that perceptions about the etiology and prognosis of asthma drive stigma and behaviors around social status, employability, denial of asthma diagnosis, refusal to seek care, and lack of treatment or inappropriate treatment. Healthcare systems in SSA are not prepared to diagnose and manage asthma and when combined with patients’ reluctance to seek care or disclose complete histories, which leads to high rate of exacerbations and death. The authors did a nice job of placing these findings within the context of research on other chronic diseases such as HIV and TB and identified several potential solutions that must occur across multiple levels in order to reduce stigma, enhance diagnosis and access to effective medications, and reduce morbidity and mortality. There was also an acceptable discussion of study limitations. All of the concerns raised during the prior review were adequately addressed.

Reviewer #2: The authors have made satisfactory amendments to the manuscript in response to my previous comments.

7. PLOS authors have the option to publish the peer review history of their article (what does this mean? ). If published, this will include your full peer review and any attached files.

**Do you want your identity to be public for this peer review?** For information about this choice, including consent withdrawal, please see our Privacy Policy .

Reviewer #1: No

Reviewer #2: No
